# Comparative Analysis of the NDVI and NGBVI as Indicators of the Protective Effect of Beneficial Bacteria in Conditions of Biotic Stress

**DOI:** 10.3390/plants11070932

**Published:** 2022-03-30

**Authors:** Nallely Solano-Alvarez, Juan Antonio Valencia-Hernández, Santiago Vergara-Pineda, Jesús Roberto Millán-Almaraz, Irineo Torres-Pacheco, Ramón Gerardo Guevara-González

**Affiliations:** 1C.A. Ingeniería de Biosistemas, Facultad de Ingeniería, Campus Amazcala, Universidad Autónoma de Querétaro, Carretera Chichimequillas s/n km1, El Marques 76265, Querétaro, Mexico; nallely.solanoa@gmail.com (N.S.-A.); juan.valencia@uaq.mx (J.A.V.-H.); irineo.torres@uaq.mx (I.T.-P.); 2Ingeniería Agroquímica, Facultad de Química, Universidad Autónoma de Querétaro, Cerro de las Campanas s/n Col. las Campanas, Santiago de Querétaro 76010, Querétaro, Mexico; 3Horticultura Ambiental, Facultad de Ciencias Naturales, Universidad Autónoma de Querétaro, Campus Juriquilla, Av. de las Ciencias s/n Juriquilla, Querétaro 76230, Querétaro, Mexico; santiago.vergara@uaq.mx; 4Facultad de Ciencias Fisico Matemáticas, Universidad Autónoma de Sinaloa, Av. de las Américas y Blvd. Universitarios, Cd. Universitaria, Culiacán 80000, Sinaloa, Mexico

**Keywords:** RGB camera, hyper-spectral image, vegetation index, PGPB, *Clavibacter michiganensis*

## Abstract

Precision agriculture has the objective of improving agricultural yields and minimizing costs by assisting management with the use of sensors, remote sensing, and information technologies. There are several approaches to improving crop yields where remote sensing has proven to be an important methodology to determine agricultural maps to show surface differences which may be associated with many phenomena. Remote sensing utilizes a wide variety of image sensors that range from common RGB cameras to sophisticated, hyper-spectral image cameras which acquire images from outside the visible electromagnetic spectrum. The NDVI and NGBVI are computer vision vegetation index algorithms that perform operations from color masks such as red, green, and blue from RGB cameras and hyper-spectral masks such as near-infrared (NIR) to highlight surface differences in the image to detect crop anomalies. The aim of the present study was to determine the relationship of NDVI and NGBVI as plant health indicators in tomato plants (*Solanum lycopersicum*) treated with the beneficial bacteria *Bacillus cereus*-Amazcala (*B. c*-A) as a protective agent to cope with *Clavibacter michiganensis subsp. michiganensis* (*Cmm*) infections. The results showed that in the presence of *B. c*-A after infection with *Cmm*, NDVI and NGBVI can be used as markers of plant weight and the activation of the enzymatic activities related to plant defense induction.

## 1. Introduction

Precision agriculture constitutes a research area with the objective of improving agricultural yield and minimizing costs by assisting management with the use of sensors, remote sensing, and information technologies [1]. Therefore, precision agriculture requires a large amount of input data on crop conditions to utilize innovative analysis tools [2]. There are several approaches to improving crop yield where remote sensing has proven to be an important methodology to determine agricultural maps to show surface differences that may be associated with many phenomena. Furthermore, agricultural surface maps can be built from geo-referenced satellites, unmanned aerial vehicles (UAVs) and rover images that enable the building of larger maps from many images [3]. On the other hand, remote sensing utilizes a wide variety of image sensors that comes from common red, green, and blue (RGB) cameras and sophisticated hyper-spectral image cameras that acquire images from outside the visible electromagnetic spectrum such as infrared, ultraviolet and thermography [4]. Furthermore, vegetation indexes (VIs) are computer vision algorithms that perform operations between color masks such as from RGB cameras and hyper-spectral masks such as near-infrared (NIR) from specific band cameras with the objective of highlighting important information and to suppress environment interference in plant images to detect, quantify or identify crop anomalies [4,5,6]. Thus, VIs provides useful information about crop health such as the soil nutrient content, water content, temperature maps, indirect photosynthetic activity, and many stress responses. One of the main trends in modern crop production is the use of plant protection agents and technologies facilitating the effective control of harmful organisms and contributing to the reduction in chemicals. The optical characteristics of crop phytosanitary states can be determined by NIR-related VIs such as the NDVI (Normalized Difference Vegetation Index), which serves as a relative indicator of the amount of photosynthetically active biomass and can accurately determine changes in crop states [7]. The NDVI is probably the most utilized and well-known VI, and it consists of a linear algebraic relation between red and NIR radiation which is measured by costly and specialized multi-sensor cameras. Therefore, it is widely utilized to detect plant anomalies such as nutrition deficiencies, indirect photosynthesis estimation, and any biotic and abiotic stress condition that can be expressed as a change in chlorophyll fluorescence emission [5]. Moreover, some RGB-based VIs is currently being utilized in precision agriculture as useful remote sensing information sources [8]. However, it is important to mention that RGB cameras are far cheaper than hyper-spectral cameras such as pure NIR, NGB (NIR-GREEN-BLUE) or multi sensor cameras, and the second main advantage of RGB-based VIs is that RGB cameras are found everywhere in devices such as UAV cameras, smartphones, and professional cameras, to name but a few [9]. The Normalized Green–Blue Vegetation Index (NGBVI) is one of the previously mentioned RGB-based VIs that was proposed by the authors, and it consists of a normalized subtraction of green minus blue masks that has proven to be useful to highlight chlorosis and necrosis damage over green plants (patent pending). Thus, NDVI and NGBVI were tested in this study to determine their usefulness for providing information about plant–pathogen relationships.

Developing sustainable and ecological methods to increase agricultural productivity without increasing the area of cultivated soil has become a critical issue for ensuring food safety. Among the various tools used to increase productivity in agriculture, the use of plant-growth-promoting bacteria (PGPB) has great potential [10]. PGPB belong to a useful and heterogeneous group of microorganisms that can be found in soil, rhizosphere, root surfaces or plant tissues, and are able to enhance plant growth and protect plants from stress conditions [11]. The technology based on suitable levels of fertilization combined with PGPB inoculation is a good strategy to reduce the use of chemical fertilizers and their environmental impacts [12]. Due to their intimate relationship with host plants, PGPBs can be explored as suitable candidates for the development of bioinoculants; however, the intricacies of host–microorganism interactions are yet to be unraveled [13]. The emergence or re-emergence and rapid spread of bacterial plant diseases are global biothreats to crop biosecurity. Increased prevalence and the rapid spread of phytobacterial diseases are facilitated by environmental changes, increased international trade and immigration, as well as the emergence of new virulence traits [14]. *Clavibacter michiganensis* subsp. *michiganensis* (*Cmm*) is a phytopathogenic bacteria causing bacterial wilt disease and severe yield losses in tomatoes and other solanaceous vegetables [15]. *Cmm* invades plant xylem vessels, causing canker on the stem, wilt and discoloration of the leaves and petioles, and lesions in the fruit [16]. The pathogen can be spread by seed and agronomic practices [17]. The usage of biocontrol agents and resistant cultivars might be alternative solutions to control the pathogen, but these methods have not yet been implemented in the field [18]. There is evidence that plant signaling pathways regulating the development of protective responses to stresses are the key targets of PGPB [19]. Pathogens cause oxidative stress in plants due to the excessive generation of reactive oxygen species (ROS); to date, extensive information has been accumulated on the ability of some PGPB, including *Bacillus subtilis* and *Bacillus cereus*, to trigger the antioxidant immune system (AOS), which increases tolerance in the associated host plant [20]. In tomato plants, it has been found that the application of *B. cereus* strain Amazcala (*B. c*-A) increases enzyme activities related to stress response and defense, and induces the innate immune system increasing tolerance to diseases caused by several pathogens, which are correlated with *Cmm* protection [15]. Based on the aforementioned information, the aim of the present study was to determine the relationship of NDVI and NGBVI as plant health indicators in tomato plant (*Solanum lycopersicum*) cultivations treated with *B. c*-A as a protective agent to cope with *Cmm* infections.

## 2. Results

### 2.1. Vegetation Index Results

The computer vision process of this project consisted of applying the NDVI algorithm to NGB images and the NGBVI algorithm to RGB images to obtain grayscale images, as shown in Figure 1. As such, twelve grayscale images are shown Figure 1, where Vis are on the rows and time-lapses per day are on the columns.

Notably, NGBVI images are more intense on the chlorotic areas of the analyzed plants. On the other hand, NDVI images are more intense on the high photosynthesis areas of the plants. Furthermore, a quantitative analysis was carried out to determine the individual properties of each of the analyzed plant specimens, which were labeled from 1 to 6 (plants inoculated with *B. c*-A.) and from A to F (plants not inoculated with *B. c*-A), as previously mentioned. Consequently, new results were obtained, showing the individual properties of the plants, as presented in Table 1.

### 2.2. Enzymatic Activities and Plant Performance

The results showed that before the infection with *Cmm*, the characteristics of plants, such as weight and height, did not exhibit a significant difference between treatments with or without the inoculation of beneficial bacteria *B. c*-A; however, 15 days after the infection with *Cmm*, significant differences in plant weight and height appeared with the beneficial bacteria in comparison with the controls (Figure 2a,b). Regarding the enzymatic activities, PAL and CAT activities before *Cmm* infection increased in treatments without beneficial bacteria *B. c*-A, but after infection, the treatments without beneficial bacteria decreased and treatments with beneficial bacteria increased (Figure 2c,d). SOD activity did not display significant differences (Figure 2e).

### 2.3. Analysis of NDVI and NGBVI

When analyzing the behavior of the NDVI before and after infection, data showed that in treatments without the inoculation of beneficial bacterial, the NDVI was significantly lower than treatments with beneficial bacteria. After infection, both treatments increased, the highest being in treatment without beneficial bacteria *B. c*-A compared with treatment with *B. c*-A (Figure 3a). NGBVI exhibited a different behavior from NDVI, after infection in both treatments with and without beneficial bacteria *B. c*-A. NGBVI increased in comparison with treatments before infection with *Cmm* (Figure 3b). It is worth mentioning that both indexes displayed significant higher measurements after *Cmm* infection treatments (Figure 3a,b).

### 2.4. Correlation Analysis between the Variables and Both Indices

A correlation matrix was constructed between all the variables: PAL, CAT, SOD activities, weight, height and NDVI and NGBVI (data not shown)—only the correlation values between VIs and the rest of the variables are given in Table 2. This study had two time points, before and 15 days after infection with *Cmm*, and in two different treatments, with or without the inoculation of beneficial bacteria *B. c*-A. It was observed that the NDVI has a negative correlation with PAL activity, weight, and height before infection without *B. c*-A; and after infection had a positive correlation with CAT and SOD activity. On the other hand, NGBVI had a positive correlation with CAT, SOD activities and weight in the presence of *B. c*-A before *Cmm* infection, but after the infection, these positive correlations were only apparent with PAL and CAT activities and appeared as a negative correlation with SOD activity and weight with the inoculation of *B. c*-A (Table 2).

## 3. Discussion

The results showed that PAL and CAT activity after infection with *Cmm* significantly increased in treatments with the inoculation of beneficial bacteria *B. c*-A, as reported by other studies, where the presence of beneficial bacteria such as many species of *Bacillus* increased biochemical and molecular variables were associated with oxidative stress responses [15,21]. It has been observed that the preventive inoculation of a beneficial microorganism reduces the incidence and severity of an infection by some phytopathogens, increasing total protein and CAT activity [22]. It has also been found that some bacteria provide protection against *Cmm* by altering the activity of PAL and total phenolics, decreasing the browning of vascular bundles and the yellowing of leaf [23]. Although the beneficial activity of *B. c*-A results shown in this study were only for the plant–pathogen interactions evaluated, it would be interesting to investigate whether the beneficial effects of *B. c*-A can also be extended to other plant species and pathosystems. The plant-growth-promoting activity of several bacteria on different crops has already been reported in grasses, *Fabaceae*, and some weeds [15]. In this study, the results of enzymatic activity and plant weight were correlated with NDVI and NGBVI, such as the relationship of NDVI and deficit irrigation [24] or the Standardized Precipitation and Evapotranspiration Index [25]. In order to develop a better understanding of the correlation results, it is important to remember that NDVI represents an indirect estimation of photosynthetic activity, and high NDVI values are related to high chlorophyll fluorescence due to NIR emissions. On the other hand, NGBVI is an RGB index where green healthy plant surfaces have a low pixel value and chlorotic–necrotic surfaces have high pixel values. Furthermore, it was observed that NDVI had a negative correlation with plant weight, height and PAL activity before the infection with *Cmm*. However, a positive correlation of NDVI with CAT and SOD activities could be observed after infection with *Cmm* without beneficial bacteria; a similar relationship has been reported in red onion with CAT activity [26] and in common bean with Water Index, GPX, SOD and enzymes [27]

On the other hand, the NGBVI had a positive correlation with CAT, SOD activities and plant weight with the inoculation of *B. c*-A, before infection with *Cmm*; however, after infection, it displayed a positive correlation with PAL and CAT activities and a negative correlation with SOD and weight. Taken together, these results suggest that the NDVI and NGBVI could be used to analyze the effect of beneficial bacteria before and after an infection.

As previously mentioned, the present study found that both NDVI and NGBVI were able to show similar significant changes in plants before and after *Cmm* infections (Figure 4 and Figure 5). Moreover, relationships between both VIs and plant enzymatic activities related to stress responses and morphological variables in the different treatments evaluated were also detected. However, it is notable that RGB-based VIs such as NGBVI are cheaper than NIR-based indexes such as NDVI, inasmuch as almost any RGB camera, which may be professional, can be used, such as the smartphone-based RGB camera that was utilized in this study.

The findings showed that NGBVI and NDVI have a similar performance in detect relationships with the enzymatic activity of tomato plants under biotic stress by *Cmm* infections; a lower cost camera can be used instead of an expensive NDVI camera to detect the presence of phytopathogenic bacteria. If we use a common camera to detect NGBVI, faster software can be used, and it could be implemented in a smartphone in the near future. In conclusion, both NDVI and NGBVI displayed significant relationship as plant health indicators in the interaction of tomato plants (*Solanum lycopersicum*) treated with the beneficial bacteria *Bacillus cereus*-Amazcala (*B. c*-A) as a protective agent to cope with *Clavibacter michiganensis subsp. michiganensis* (*Cmm*) infections.

## 4. Materials and Methods

### 4.1. Plant Materials and Experimental Design

The experiment was carried in a 50 m^2^ greenhouse. The average temperature of the greenhouse was 29 °C ± 3 °C and the relative humidity was 70%; the photoperiod corresponded to natural light of 12:12 light–dark cycles. Plants were irrigated daily to field capacity (20 mL per day) and the nutrient solution used was Steiner [28]. The tomato plants were grown in pots with a substrate (a 3:1 mix of peat moss/vermiculite) in two different treatments with the inoculation of *B. c*-A and without bacteria, using six plants per treatment. At 30 days old, the height of the plants was monitored with a tape measure, and the total fresh weight was obtained by cutting the plants from the base.

### 4.2. Preparation of Inoculants and Infection

*B. c*-A and *Cmm* were grown in LB (Luria Bertani) broth with constant agitation of 180 rpm at 30 °C for 48 h. At the end of the logarithmic phase, the bacterial culture was centrifuged at 300 rpm for 3 min and the supernatant was discarded; the pellet was resuspended in sterile water. The aqueous suspension was diluted until it reached a concentration of 1 × 10^8^ CFU mL^−1^. The seeds were immersed in 1 × 10^8^ CFU mL^−1^ culture of *B. c*-A or sterile LB broth for 24 h, depending on the treatment. After that time, the seeds were placed in a seedling. Thirty-day-old plants were transplanted in pots and treatment with *B. c*-A was inoculated, applying 1 mL per kg of substrate of an inoculum grown in LB (1 × 10^8^ CFU mL^−1^). Three days later, after the transplant and *B. c*-A application, the *Cmm* strain (AcR42, kindly donated by Dr. Angel Alpuche-Solis, IPICyT, México) was inoculated by puncturing the stem with a bacterial suspension grown in liquid LB (1 × 10^8^ CFU mL^−1^) in all the plants in both treatments, according to [15]. After 15 days of infection with *Cmm*, the apical leaves were recollected and immersed in liquid nitrogen for subsequent lyophilization.

### 4.3. Stress Response-Antioxidant Enzyme Activities Measurements

Total soluble proteins (TSPs) were extracted from lyophilized leaves (50 mg) ground with 1.5 mL extraction buffer containing 0.05 M phosphate buffer (pH 7.8) and 0.1 mM Na_2_EDTA. The samples were centrifuged at 12,000 rpm for 15 min at 4 °C. Protein concentrations were determined using the Bradford Reagent (Sigma, Ronkonkoma, NY, USA) and bovine serum albumin (Sigma, USA) as the standard. The supernatant obtained for the TSP assay was used to quantify catalase (CAT), phenylalanine ammonia lyase (PAL), and superoxide dismutase (SOD) activities. CAT activity was assayed by following the initial rate of H_2_O_2_ degradation for 120 s monitored at 240 nm; enzyme activity was determined with some modifications [29]. PAL was determined by spectrophotometry at 290 nm, quantifying the cinnamic acid formed from the catalysis of L-phenylalanine [30]. SOD activity was determined by spectrophotometry at 560 nm by measuring the photochemical nitroblue tetrazolium (NBT) reduction [31].

### 4.4. RGB and NIR Image Acquisition

Taking photographs and registries on day one, when plants had completed 30 days of age; day two, with the inoculation with *B. c*-A; day three, with infection with *Cmm*; and day six, after 15 days of infection, with a common RGB camera and a second NIR camera, generated the image database of this experiment. In order to acquire RGB images, a Samsung Galaxy S7 Edge smartphone camera was utilized as the RGB image source at a 4032 × 3024 pixel resolution with 4:3 aspect ratio. The NIR acquisition process was performed by utilizing a modified Canon PowerShot A480 camera at a 3648 × 2736 pixel resolution with the IR-blocking filter removed and replaced with a standard blue filter for NDVI applications obtained from Public Lab to implement an NIR–Green–Blue (NGB) camera using the single-sensor NDVI camera technique where a red mask is replaced with an NIR mask in the camera sensor [32]. Consequently, a dataset with 7 RGB photographs and 7 NGB photographs was generated in order to observe any minor change across the time-lapse of one week, as shown in Figure 4, where it is observed that plants A to F were not inoculated with *B. c*-A, and plants 1 to 6 were inoculated with *B. c*-A.

### 4.5. NDVI Analysis

NDVI is by far the most utilized VI for precision agriculture applications, and it has proven to be useful to visualize indirect photosynthetic activity over fields that are related to biotic and abiotic stress conditions. To calculate the NDVI, linear algebra operations need to be performed between NIR and RED masks according to Equation (1) [4]. As previously mentioned, this study utilized a single-camera NDVI technique where an NGB camera with a blue filter was utilized and the NDVI was calculated using Equation (2), where NIR came from the red mask place in the NGB image, and due to the absence of the RED mask, the BLUE mask was utilized instead of the RED mask to obtain an NDVI1-type result.
(1)$$\mathrm{NDVI}=\frac{\mathrm{NIR}-\mathrm{RED}}{\mathrm{NIR}+\mathrm{RED}}$$
(2)$$\mathrm{NDVI}1=\frac{\mathrm{NIR}-\mathrm{BLUE}}{\mathrm{NIR}+\mathrm{BLUE}}$$
where NIR is near-infrared, RED is the red light, and BLUE is the blue light.

### 4.6. NGBVI Analysis

NGBVI is a novel RGB-based VI that was recently proposed by the authors of the present study, and it focuses on highlighting chlorosis and necrosis surfaces over green plants with a good capability for ignoring non-vegetation surfaces in the RGB image. Furthermore, the NGBVI can be calculated using Equation (3), where the subtraction of GREEN minus BLUE is divided by the maximum pixel value of the same subtraction. Therefore, a grayscale NGBVI is obtained as a result, and it can be converted to a color scale such as Jet (i.e., the color ‘jet’, also called jet black), CIELab (The CIE 1976 L*a*b* color space), HSV (hue saturation value), to name but a few if the user wants to highlight image differences in a colorful way.
(3)$$\mathrm{NGBVI}=\frac{\mathrm{GREEN}-\mathrm{BLUE}}{max\left(\mathrm{GREEN}-\mathrm{BLUE}\right)}$$

### 4.7. Quantification Algorithm

As mentioned above, six plants were selected from each of the two statistical groups for quantification purposes. In order to extract plant features per each specimen, a sub-mask was analyzed separately for each plant under study with the objective of quantifying specific feature values from each plant. Furthermore, a quantification algorithm was applied to each plant mask, as observed in Figure 5, where the plant sub-mask was extracted and further binarized to build the new BW mask. Finally, the quantification algorithm was calculated according to Equation (4), where NBP is the number of non-black pixels, *N* is the *x*-size, *M* is the *y*-size, and all the non-black pixel values from IMG are accumulated to obtain a quantified result, *Q*, per plant.
(4)$$Q=\frac{1}{\mathrm{NBP}}\overset{N}{\underset{x=1}{\sum }}\times \overset{M}{\underset{y=1}{\sum }}\times \mathrm{IMG}\left(x,y\right)$$

### 4.8. Statical Analysis

To assess the enzyme activities, plant height and weight measurements, a normality test of data (Anderson–Darling test, *p* = 0.05) was carried out. All the results obtained were expressed by the means plus/minus their standard errors. The differences between treatments were determined by one-way ANOVA, and differences between the means were determined by Tukey’s test (*p* = 0.05) using GraphPad PRISM version 8 software.

### 4.9. Ethical Statement

The authors declare that they have no conflicts of interest. This manuscript does not contain any studies with human or animal participants performed by any of the authors, and this project was approved by the ethics committee from School of Engineering from Autonomous University of Querétaro (Querétaro, México) with approval reference number CEAIFI-100-2018-TP.

## Figures and Tables

**Figure 1 plants-11-00932-f001:**
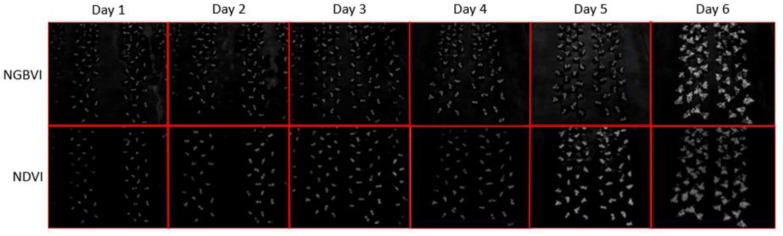
Block diagram for the overall quantification algorithm.

**Figure 2 plants-11-00932-f002:**
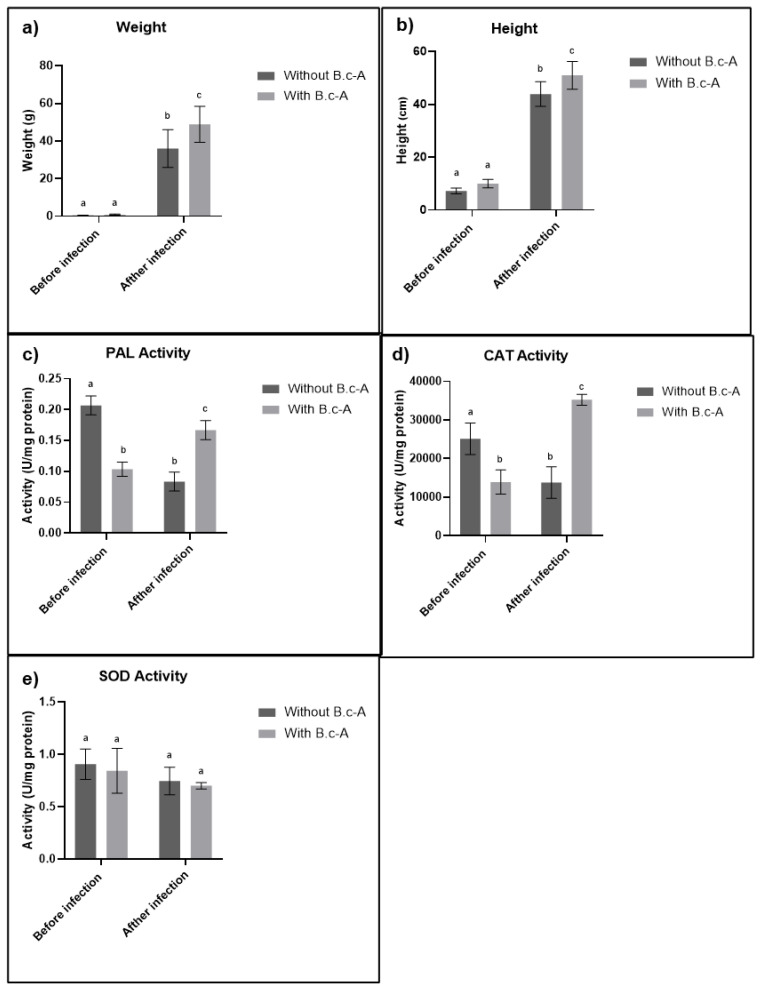
Weight (**a**), height (**b**), and enzymatic activity (**c**–**e**), of 30-day-old plants with and without the inoculation of *B. c*-A, before and after infection with *Cmm* (equal letters indicate that there is no statistically representative difference).

**Figure 3 plants-11-00932-f003:**
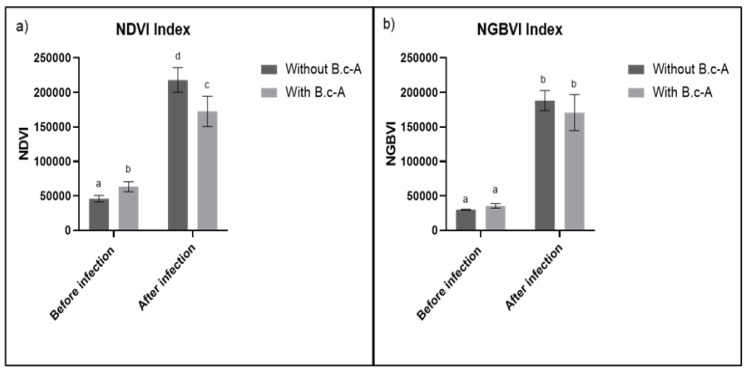
(**a**) NDVI values before and after infection with Cmm, (**b**) NGBVI before and after infection with Cmm. Equal letters indicate that there are no statistically representative differences.

**Figure 4 plants-11-00932-f004:**
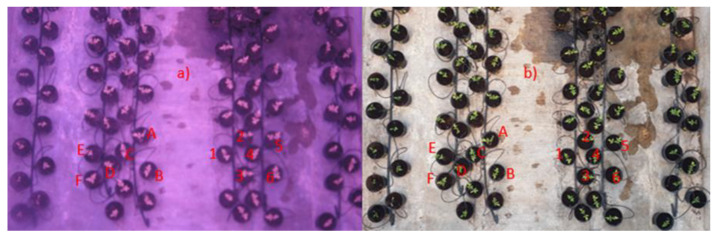
Example of acquired images with both cameras aimed at the same objective. (**a**) NGB camera image. (**b**) RGB camera image. Simbology: plants A to F were not inoculated with *B. c*-A, and plants 1 to 6 were inoculated with *B. c*-A.

**Figure 5 plants-11-00932-f005:**
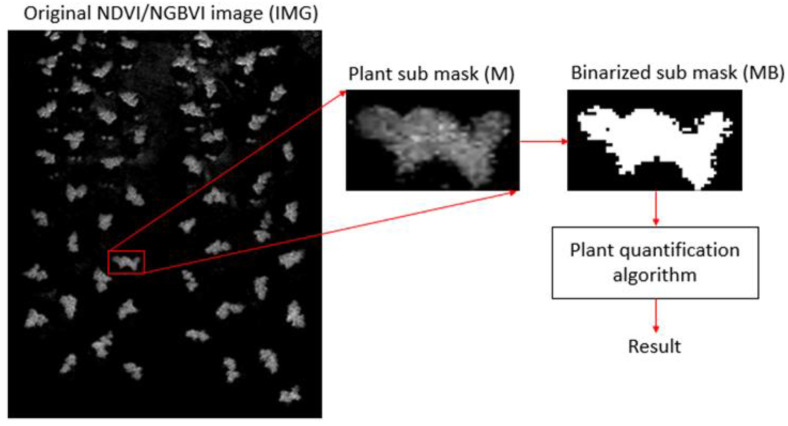
Block diagram for the overall quantification algorithm.

**Table 1 plants-11-00932-t001:** NDVI and NGBVI quantification analysis results per plant.

	Plant	Day 1	Day 2	Day 3	Day 4	Day 5	Day 6
NDVI	A	13,512	38,696	26,222	34,942	75,066	163,026
	B	15,951	46,849	29,788	37,027	72,366	197,455
	C	18,516	48,343	33,660	39,114	77,139	228,055
	D	15,046	44,383	33,937	47,203	93,095	228,856
	E	7168	25,590	24,373	15,259	59,555	103,185
	F	13,304	51,194	40,365	45,694	88,859	166,371
	1	14,362	44,403	25,367	25,093	54,217	147,193
	2	16,853	54,138	30,826	39,690	69,922	136,349
	3	14,487	63,562	33,203	40,506	86,770	184,440
	4	15,352	58,641	30,771	34,623	72,949	88,062
	5	16,464	66,804	45,331	53,131	98,997	185,975
	6	22,477	73,057	49,660	51,721	119,307	264,877
NGBVI	A	26,401	29,288	38,355	43,606	39,701	199,574
	B	27,354	29,609	30,952	39,641	28,151	193,025
	C	28,271	31,934	38,610	42,215	34,252	242,995
	D	22,297	27,658	37,635	40,393	36,940	227,300
	E	18,437	21,000	25,314	31,615	38,491	151,406
	F	24,085	30,565	34,473	45,871	37,396	171,875
	1	24,323	25,331	26,133	31,398	23,493	152,722
	2	25,887	26,919	33,599	35,851	27,261	158,895
	3	24,561	32,090	38,172	47,092	36,211	200,734
	4	21,494	25,647	27,962	33,164	22,571	95,647
	5	27,073	35,677	41,425	52,773	43,204	235,303
	6	32,647	38,542	41,756	49,522	39,076	259,662

**Table 2 plants-11-00932-t002:** Correlation values of the NDVI and NGBVI before and after infection with *Cmm*.

	Before Infection with *Cmm*									
	Without Inoculation of *B. c*-A					With Inoculation of *B. c*-A				
	PAL	CAT	SOD	Weight	Height	PAL	CAT	SOD	Weight	Height
NDVI	−0.95	−0.09	0.73	−0.85	−0.93	−0.14	−0.94	−0.60	−0.21	0.37
NGBVI	0.58	0.66	−0.19	0.38	0.97	0.69	0.96	0.96	0.75	0.24
	After infection with *Cmm*									
	Without inoculation of *B. c*-A					With inoculation of *B. c*-A				
	PAL	CAT	SOD	Weight	Height	PAL	CAT	SOD	Weight	Height
NDVI	−0.70	0.94	0.91	−0.96	−0.10	0.70	−0.33	−0.09	−0.05	0.99
NGBVI	−0.85	0.52	−0.23	−0.47	−0.99	0.85	0.86	−0.99	−0.99	0.05

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
