# Peer review of "Comparative Analysis of the NDVI and NGBVI as Indicators of the Protective Effect of Beneficial Bacteria in Conditions of Biotic Stress"

_plants, 2022, doi:10.3390/plants11070932_

Round 1
Reviewer 1 Report
The originality of this paper is to use low cost RGB cameras in the domain of plant health with the idea of relating these measurements to well known NDVI. However the RGB measurements are compared with a modified camera that allows near infrared detection. This approach in my opinion is fundamentally incorrect as the these modified cameras have large and overlapping spectral responses. In other words the 1) the modified camera does not measure true NDVI. 2) There is inherent correlation between the three (large and overlapping bands). As such is it completely understandable that the two methods detect similar difference. This does not however answer the over whether NDVI and NGBVI can be compared. As such I do not think that the paper can be accepted.
In terms of structure having materials and methods at the end is strange. There are several English and figure reference faults.
Author Response
Dear Reviewer
All your kind suggestions to my submission are included in purple colour in the revised manuscript attached.
Best Regards,
Dr. Ramon G. Guevara-Gonzalez
Corresponding author

Reviewer 2 Report
The manuscrip entitled "Comparative analysis of the NDVI and NGBVI indexes as indicators of the protective effect of a PGPB in conditions of biotic 3 stress" is globally well written and presents the capacity of NDVI and NGBVI indexes to detect the action of beneficial bacteria.
The introduction is complete with few minor errors, see in the document attached.
The results could be improved in the form.
The material and methods should be improved by clarifying the date/stage of the measurements along the description of the different methods and results.
The discussion is good.
Please find more comments on the document.
The format of the references should be completely review.

Author Response
Dear Reviewer
All your kind suggestions has been corrected in the revised manuscript and are highlighted in green color.
Best Regards,
Dr. Ramon G. Guevara-Gonzalez
Corresponding author

Reviewer 3 Report
Manuscript ID: plants-1568989
Type of manuscript: Article
Title: Comparative analysis of the NDVI and NGBVI indexes as indicators of the protective effect of a PGPB in conditions of biotic stress
In present study authors assess the relationship of NDVI (Normalized Difference Vegetation Index) and NGBVI (Normalized Green-Blue Vegetation Index) indexes as plant health indicators in tomato (Solanum lycopersicum) cultivations treated with B.c-A (Bacillus cereus-Amazcala) as a protective agent to cope Cmm (Clavibacter michiganensis subsp. michiganensis) infections. Present study found that both NDVI and NGBVI indexes were able to show similar significant changes in plants before and after Cmm infections. Relationships between both vegetation indexes and plant enzymatic activities related to stress response and morphological variables in the different treatments were also detected.
Comments and Suggestions for Authors:
In my opinion, there are some loopholes in the data processing and comparison of results with literature data.
Abbreviations should not be used in the Title of this manuscript and in Abstract.
Overall, in the text of the manuscript there are too many abbreviations for the reader to be able to orient themselves in the continuous reading of the text.
Line 54: [4], [5] (delete), correctly: [4,5].
Line 112: infections.. (duplicate dot)
Line 118: where VIs is (delete) are on the rows and time-lapse per day is on the columns.
Line 117: Figure 3: incorrect numbering
Line 233: in all the plants in both treatments, according to Solano-Alvarez et al 2021 (delete), correctly: according to [24].
Lines: 275–276: explain the abbreviations in the formulas
Line 284: Explain the abbreviations: such as Jet, CIELab, HSV
Line 296: Finally, the quantification algorithm is calculated according to equation (D) where NBP are (delete) is the number
Figure 2: weight (g gr )
Discussion:
Overall a weak discussion, only 4 papers are cited.
Lines 193–198: “On the other hand, NGBVI index had a positive correlation with CAT, SOD activities and plant weight with inoculation of B. c-A, before infection of Cmm, but after infection it was displayed a positive correlation with PAL and CAT activities and a negative correlation with SOD and weight. Taking together, these results suggested that NDVI and NGBVI index could be used to analyze the effect of beneficial bacterial before and after an infection” – Justify importance of the relationships between the variables of both indices in comparison with the literature.
Line 177: incorrect citation: [20] [14], correctly: [14,20]
Lines: 206–210: Incomprehensible conclusion of the sentence, re-stylize:
Consequently, NGBVI showed a similar performance than like NDVI to detect significant correlations with enzymatic activity of tomato plants under biotic stress by Cmm infections but utilizing a lower cost camera, which may result useful for faster hardware, and software implementations that may be implemented in a smartphone in a near future.highlighted.
In total, the authors cite only 27 pieces of literature.
In discussion section of the manuscript it is necessary to compare thoroughly the obtained results with the literature data.
The Section References is not cited according to the guidelines of the journal “Plants”.
It is necessary to renumber the literature throughout the manuscript.
Line 389: (27. NASA, «The AREN Proyect,» Spinoff, 2018) Is it a complete citation ?
More general considerations need to be added in section discussion.
The paper was not written in standard, grammatically correct English.
The conclusion of this article is missing.
What is the practical significance of the results in relation to the environment? Add considerations.
I advise the manuscript to be adjusted, including formal as well as content page. Large corrections needed.
Date of this review:
January 18, 2022

Author Response
Dear Reviewer
Thanks for your kind comments. All your suggestions have been included in the revised manuscript and are highligthed in yellow color.
Best Regards,
Dr. Ramon G. Guevara-Gonzalez
Corresponding author

Reviewer 4 Report
The manuscript plants-1568989 investigated the NDVI and NGBVI suitability as plant health indicators in tomato plants (Solanum lycopersicum) treated with the biocontrol bacteria Bacillus cereus-Amazcala against Clavibacter michiganensis subsp. michiganensis infections.
The manuscript topic is of great interest, the methods applied are appropriate, and the data handling is suitable. The findings reported are interesting, and the authors correctly discussed the results from the perspective of the working hypothesis and literature on the main topic. However, the manuscript presentation should be improved only for some minor issues. Please see specific comments below.
- I would expand the discussion in lines 178-186. Even if the goal of the study was not to evaluate the aspects related to the PGPB effects, this part of the discussion seems a little more imbalanced than the Introduction and Results sections.
- Revise the italics for scientific names throughout the manuscript.
Author Response
Dear reviewer
Thanks a lot for your kind comments on my submission. Attached I am sending a corrected version of the manuscript according to your comments and suggestions. These corrections are highlighted in yellow in the revised text.
Respectfully,
Dr. Ramon G. Guevara-Gonzalez

Round 2
Reviewer 1 Report
Very little has changed from the first version. Again it is normal that results are correlated as the authors NDVI camera is not a true band spectral camera. This is a major flaw. I see no way round this and the authors do not address the question.
Still no conclusion !
Sorry I suggestion that the article is rejected.
Author Response
Thanks for your comments.
Attached file have the address to the comments.
Best regards,
Dr. Ramon G. Guevara-Gonzalez

Reviewer 3 Report
Accept in present form.
Author Response
Dear Reviewer
Thanks for your comments.
Respectfully,
Dr. Ramon G. Guevara-Gonzalez
Corresponding author
